# OpenReview forum: "JAILJUDGE: A Comprehensive Jailbreak Judge Benchmark with Multi-Agent Enhanced Explanation Evaluation Framework"
_ICLR.cc/2025/Conference — Submitted to ICLR 2025_

### Official Review · Reviewer_Ascp · 2024-10-30

**Soundness:** 3
**Presentation:** 2
**Contribution:** 2
**Rating:** 5
**Confidence:** 4

**Summary:**

This paper proposes a comprehensive jailbreak judge benchmark with a multi-agent enhanced explanation evaluation framework, motivated by the challenge of existing evaluation methods lacking explainability and generalization to complex scenarios. To address the previous issues, the JAILJUDGE dataset comprises over 35k+ instruction-tune training data with reasoning explainability and test data with 4.5k+ broad risk scenarios and 6k+ multilingual scenarios. To provide reasoning explanations and fine-grained evaluations, this work also proposes a multi-agent jailbreak judge framework to enhance the evaluation quality with explicit interpretation. With the instruction-tuned model, an attacker-agnostic enhancer and a safety moderation defense method are designed based on the reliable evaluation benchmark. Comprehensive experiments are presented to evaluate the effectiveness of the proposed methods.

**Strengths:**

1. This paper aims to address the fundamental challenge of jailbreak evaluation, e.g., the unreliable evaluation without explainability, and under-preformed generalization to complex scenarios, which is important to the research development towards the safe usage of large language models (LLMs).
2. The benchmark incorporates the 10 multilingual languages to enable the evaluation of the multilingual scenarios, and also provide various high-quality labeled data for both instruction-tuning and broad risk testing, it would provide more available empirical resources for the following jailbreak research to investigate the risks of LLMs in different scenarios.
3. This work designs a multi-agent judge framework that incorporates a voting mechanism to give a jailbroken score and reason during the evaluation and seems to be promising for the evaluation.
4. The proposed jailbreak attack enhancer and defense are demonstrated to be effective by comparing with different conventional baselines.

**Weaknesses:**

1. The methodology or mechanism innovation is limited although the primary area of this work is dataset and benchmarks. In general, the difference between this work from the previous jailbreak judge benchmark or methods is not significant according to the summarization of Table 1. Specifically, compared with WildGuard, JAILJUDGE incorporates multilingual languages; while at the method aspect, the differences are mainly on the open source model and data.
2. The claimed contributions are overwhelming but not clearly compared with the previous research works in each aspect (e.g., the jailbreak evaluation benchmark, the jailbreak attack, the jailbreak defense, and the explanation ability of LLM), which makes it hard for readers or reviewers to find a unique fundamental innovation of the benchmark with the proposed attack and defense methods although they are comprehensive.
3. Although the paper claimed that the previous jailbreak evaluation would induce incomplete and inaccurate assessments, there is limited empirical evidence to justify or demonstrate the challenge or issue that existed in the previous evaluation benchmarks. The lack of empirical evidence makes it hard to assess the innovation quality.
4. Except for collecting the instruction-tune training data with reasoning explainability, it is unclear how this proposal can address the previous issue of lacking a reliable explanation about jailbreak behaviors.
5. The organization and presentation of the current version are not good, although the reviewer can be impressed by various contributions to the direction of research. It could be better if the author could significantly improve the presentation with clear problem illustration and state the rationality of the proposed evaluation framework.

**Questions:**

1. What is the difference between the claimed "multi-agent" and multi-inference using LLMs? It would be better if the authors could provide a clearer definition or explanation about the concept of "multi-agent" adopted in this work, since it may be different from the concept used in deep reinforcement learning.
2. What is the implication of using evidence theory in modeling the hypothesis of whether an LLM is jailbroken or not?  Is there a significant difference from the traditional probability theory? If it is directly adopted in the judging design, what is the unique contribution in the uncertainty-aware transformation? and how to decide the hyperparameter in the transformation?

Minor comments:
1. The current version may need to change \citet to \citep to make the citations more obvious in the main text.
2. If one major contribution of this work would be the high-quality labeled dataset, it would be better to incorporate something like the datasheet as encouraged by the neurips dataset benchmark track, to give a more detailed overview of the data property and collection process.

---

### Official Review · Reviewer_81E4 · 2024-11-01

**Soundness:** 1
**Presentation:** 1
**Contribution:** 2
**Rating:** 3
**Confidence:** 4

**Summary:**

The paper proposes a dataset, a judge model, and a defense model meant to study unsafe responses by LLMs.

**Strengths:**

It is good that the paper grounded the generation of the prompts it used in a concrete policy (specifically, the MLCommons one). More papers on safety and jailbreaking should ground to a robust, specific policy definition instead of leaving definitions open at the high level.

**Weaknesses:**

I have concerns about the quality of the underlying dataset of vanilla prompts that can be addressed by more detailed discussions of quality metrics for the human annotation process and expert relabeling of conversations. Manual inspection of the anonymized dataset reveals that several prompts are not complete sentences and it is not clear that many of them would truly violate the MLCommons line. For example, some prompts solicit instructions on building a gambling website.

Any proposal of a defense model should consider false refusals, as it is easy to reject all attacks by destroying all utility of the model.

**Questions:**

Line 200: “selected to optimize coverage, diversity, and balance” Are there metrics the authors can share about these properties of the dataset?

Line 264: “We provide detailed scenarios and examples to the human annotators, allowing them to learn what constitutes a violation of these policies” Can the authors share these guidelines?

Line 270-274: Can the authors release specific quality metrics about the prompts generated by the human annotators? More specific numbers on the quality assurance process here would be very helpful.

Lines 958/959: The authors mention that they “ncorporate prompts from various datasets” Can the authors provide better provenance of each prompt? Are they sure they have the rights to use such data?

Can the authors report the false refusal rate that the GuardShield model achieves relative to the other defense methods they benchmarked against?

---

### Official Review · Reviewer_jar5 · 2024-11-02

**Soundness:** 3
**Presentation:** 3
**Contribution:** 3
**Rating:** 6
**Confidence:** 4

**Summary:**

This paper presents JAILJUDGE, a comprehensive benchmark and evaluation framework designed to assess jailbreak vulnerabilities and defenses in LLMs, which includes a wide range of risk scenarios with complex malicious prompts (e.g., synthetic, adversarial, in-the-wild, and multi-language scenarios, etc.) along with high-quality human-annotated test datasets. To enhance interpretability, the authors propose a multi-agent judge framework, JailJudge MultiAgent, which evaluates LLM responses with fine-grained scoring (1 to 10) and provides reasoning explanations. Additionally, JAILJUDGE Guard, an end-to-end jailbreak judge model, is introduced to offer explainable evaluations without incurring API costs. The paper also demonstrates two applications: JailBoost, an attack enhancer that increases jailbreak success rates, and GuardShield, a defense mechanism that significantly improves LLM robustness against jailbreak attempts.

**Strengths:**

1. **Comprehensive Benchmark**: The paper introduces the JAILJUDGE benchmark, which covers a broad range of complex and multilingual jailbreak scenarios, addressing gaps in existing evaluation datasets with high-quality, human-annotated data.

2. **Explainability-Enhanced Multi-Agent Framework**: The multi-agent judge framework (JailJudge MultiAgent) provides reasoning explainability and interpretable, fine-grained evaluations, making the decision process clearer and more reliable.

3. **State-of-the-Art Performance**: The proposed methods, including JAILJUDGE Guard, achieve competitive results, surpassing existing baselines and demonstrating high performance across both in-distribution and out-of-distribution scenarios.

**Weaknesses:**

1. **Heavy Dependence on GPT-4 Annotations**: A significant portion of the dataset’s annotations is generated by GPT-4, which may introduce biases inherent to this model. Given this, why was GPT-4 chosen as the annotation model, and what measures were implemented to mitigate GPT-4’s biases in the annotation process?

2. **Multilingual Baseline Comparison**: While JAILJUDGE includes multilingual scenarios, and Figure 2 shows F1 scores across ten different languages, it remains unclear how its performance compares specifically in low-resource languages, where biases and vulnerabilities might differ. Adding baseline comparisons in low-resource languages could better demonstrate JAILJUDGE's effectiveness in these challenging settings.

3. **Component-Level Ablation for Multi-Agent Framework**: The ablation study in section 5.3 evaluates four configurations of the multi-agent judge framework, progressively adding components to measure their collective impact. However, the study primarily focuses on comparing the complete multi-agent framework to partially enhanced models rather than isolating and analyzing the individual contributions of each component (e.g., uncertainty-aware judging agent, reasoning-enhanced GPT-4) separately. Given this, could the authors provide an analysis of each component's contribution in isolation to clarify its unique role and necessity within the framework?

**Questions:**

See the Weaknesses above.

---

### Official Review · Reviewer_8dGw · 2024-11-04

**Soundness:** 3
**Presentation:** 3
**Contribution:** 3
**Rating:** 8
**Confidence:** 4

**Summary:**

This paper introduces JAILJUDGE, a comprehensive evaluation benchmark, and framework for assessing LLM-as-a-judge for safety settings. The authors create two main components: (1) A large benchmark dataset (JAILJUDGE) containing 35k+ training examples and 10k+ test examples spanning multiple languages and attack types, and (2) A multi-agent framework for evaluating jailbreak attempts that provides explanations and fine-grained scoring. The authors also develop JAILJUDGE Guard, an open-source model trained on their dataset that can evaluate jailbreak attempts. Experiments show their approach outperforms existing methods across multiple metrics.

**Strengths:**

-This paper is well structured and has many contributions, I commend the authors for the great effort.
- The multi-agent judge framework makes valuable methodological contributions, particularly in how it handles uncertainty through evidence theory. The use of separate judging, voting, and inference agents with explicit uncertainty modeling is an interesting idea.
- Incorporating the proposed judge into attacks and defenses is interesting
- Dataset for evaluating and training judge model is comprehensive, incorporating additional types of data such as adversarial prompts and synthetic data

**Weaknesses:**

- The OOD evaluation dataset is only OOD for language. It does not incorporate new risk categories, so there may be some potential overfitting on the risk scenarios in the training set.
- No discussion of limitations

Minor
- typo on line 087 (instruction)

**Questions:**

- Why does PAIR + JailBoost perform worse than with LLamaGuard?
- What is the difference between JailJudge+GPT4 and GPT4-Multiagent voting?
- Does the choice of the base model for JailJudge matter? Would initializing from LlamaGuard improve performance?

---

### Meta-Review · Area_Chair_r88Y · 2024-12-18

**Metareview:**

This paper proposes a comprehensive jailbreak judge benchmark with a multi-agent enhanced explanation evaluation framework, motivated by the challenge of existing evaluation methods lacking explainability and generalization to complex scenarios. To address the previous issues, the JAILJUDGE dataset comprises over 35k+ instruction-tune training data with reasoning explainability and test data with 4.5k+ broad risk scenarios and 6k+ multilingual scenarios. To provide reasoning explanations and fine-grained evaluations, this work also proposes a multi-agent jailbreak judge framework to enhance the evaluation quality with explicit interpretation. With the instruction-tuned model, an attacker-agnostic enhancer and a safety moderation defense method are designed based on the reliable evaluation benchmark. Comprehensive experiments are presented to evaluate the effectiveness of the proposed methods. However, there are several points to be further improved. For example, the methodology or mechanism innovation is limited although the primary area of this work is dataset and benchmarks. In general, the difference between this work from the previous jailbreak judge benchmark or methods is not significant according to the summarization of Table 1. Specifically, compared with WildGuard, JAILJUDGE incorporates multilingual languages; while at the method aspect, the differences are mainly on the open source model and data. The claimed contributions are overwhelming but not clearly compared with the previous research works in each aspect (e.g., the jailbreak evaluation benchmark, the jailbreak attack, the jailbreak defense, and the explanation ability of LLM), which makes it hard for readers or reviewers to find a unique fundamental innovation of the benchmark with the proposed attack and defense methods although they are comprehensive. Moreover, there are concerns about the quality of the underlying dataset of vanilla prompts that can be addressed by more detailed discussions of quality metrics for the human annotation process and expert relabeling of conversations. Manual inspection of the anonymized dataset reveals that several prompts are not complete sentences and it is not clear that many of them would truly violate the MLCommons line. For example, some prompts solicit instructions on building a gambling website. Besides, the organization and presentation of the current version are not good, although the reviewer can be impressed by various contributions to the direction of research. It could be better if the author could significantly improve the presentation with clear problem illustration and state the rationality of the proposed evaluation framework. Therefore, this paper cannot be accepted at ICLR this time, but the enhanced version is highly encouraged to submit other top-tier venues.

**Additional Comments On Reviewer Discussion:**

Reviewers keep the score after the rebuttal.

---

### Decision · Program_Chairs · 2025-01-22

Reject